# Application of Hydrological and Sediment Modeling with Limited Data in the Abbay (Upper Blue Nile) Basin, Ethiopia

Banteamlak Kase Abebe [1,2], Fasikaw Atanaw Zimale [3], Kidia Kessie Gelaye [2], Temesgen Gashaw [4], Endalkachew Goshe Dagnaw [2] and Anwar Assefa Adem [4,5,*]

1   School of Hydraulic and Water Resources Engineering, Dilla University, Dilla P.O. Box 419, Ethiopia
2   Department of Hydraulic and Water Resource Engineering, University of Gondar, Gondar P.O. Box 196, Ethiopia
3   Faculty of Civil and Water Resource Engineering, Bahir Dar Institute of Technology, Bahir Dar University, Bahir Dar P.O. Box 26, Ethiopia
4   Department of Natural Resources Management, Bahir Dar University, Bahir Dar P.O. Box 5501, Ethiopia
5   Blue Nile Water Institute, Bahir Dar University, Bahir Dar P.O. Box 26, Ethiopia
*   Correspondence: anwar.assefa@bdu.edu.et; Tel.: +251-913640284

**Abstract:** In most developing countries, biophysical data are scarce, which hinders evidence-based watershed planning and management. To use the scarce data for resource development applications, special techniques are required. Thus, the primary goal of this study was to estimate sediment yield and identify erosion hotspot areas of the Andasa watershed with limited sediment concentration records. The hydrological simulation used meteorological, hydrological, suspended sediment concentration, 12.5 m Digital Elevation Model (DEM), 250 m resolution African Soil Information Service (AfSIS) soil, and 30 m resolution land-cover data. Using the limited sediment concentration data, a sediment rating curve was developed to estimate the sediment yield from discharge. The physical-based Soil and Water Assessment Tool (SWAT) model was employed to simulate streamflow and sediment yield in a monthly time step. The result shows that SWAT predicted streamflow with a coefficient of determination ($R^2$) of 0.88 and 0.81, Nash–Sutcliffe Efficiency (NSE) of 0.88 and 0.80, and percent of bias (PBIAS) of 6.4 and 9.9 during calibration and validation periods, respectively. Similarly, during calibration and validation, the model predicted the sediment yield with $R^2$ of 0.79 and 0.71, NSE of 0.72 and 0.66, and PBIAS of 2.7 and −8.6, respectively. According to the calibrated model result in the period 1992–2020, the mean annual sediment yield of the watershed was estimated as 17.9 t ha$^{-1}$yr$^{-1}$. Spatially, around 22% of the Andassa watershed was severely eroded, and more than half of the watershed (55%) was moderately eroded. The remaining 23% of the watershed was free of erosion risk. Therefore, the findings suggests that applying the sediment rating curve equation, in conjunction with hydrological and sediment modeling, can be used to estimate sediment yield and identify erosion hotspot areas in data-scarce regions of the Upper Blue Nile Basin in particular, and the Ethiopian highlands in general with similar environmental settings.

**Keywords:** sediment rating curve; SWAT; calibration; validation; streamflow; sediment yield; data scarce region; erosion prone areas

## 1. Introduction

Soil degradation is a process by which soils' current and/or future capacity to produce goods or services is reduced [1]. Soil erosion, which causes soil degradation, is one of the most harmful processes of soil degradation and reduction in soil fertility [2]. Soil degradation is a major issue because of its negative effects on agricultural productivity, the environment, food insecurity, and quality of life [3]. According to Tamene and Vlek [4], water erosion accounts for approximately 55% of the world's 2 billion ha of degraded soils. That is, approximately 915 million ha of land was degraded due to water erosion which lowered

crop production by 2–5% per millimeter of soil loss [4]. Crop production reductions owing to previous erosion range from 2% to 40% in Africa, with a mean drop of 16% by 2020 [5].

In Ethiopia, soil productivity loss due to water indicted erosion and unsustainable land management practices is substantially affecting agricultural productivity, which accounts for 85% of the country's economy [6]. According to international standards, the average crop yield in Ethiopia is very low, owing primarily to soil fertility loss caused by erosion of topsoil [7]. Tamene et al. [8] estimated that cropland soil loss due to erosion accounts for approximately 42 tons ha$^{-1}$ yr$^{-1}$. Taddese [9] found that the 1.5 million tons of annual grain yield drop in Ethiopia was caused by the 1.5 billion tons of annual soil loss.

Soil erosion has ramifications that go beyond reduced agricultural yields. Siltation and sedimentation are two more impacts of soil erosion that affect the service life of reservoirs. A continuous supply of sediment is transported and deposited in reservoirs due to upland and river channel erosion [10]. Sediment deposition affects 1% of the global storage capacity of reservoirs on a yearly basis [11]. Reservoir storage capacity is dwindling at a significantly faster rate in some emerging countries where watershed management policies are poor. Ayele et al. [10] in Koga reservoir, Shiferaw and Abebe [12] in Abrajit reservoir, and Moges et al. [13] in Selamko and Shina reservoirs evaluated storage capacity reduction in Ethiopian reservoirs.

Records of streamflow and suspended sediment concentration that are being gathered by government organizations, academic institutions, and private people are the most crucial and vital information to comprehend the condition and trends of river water quantity and quality [14]. Insights into the patterns and variability of streamflow and sediment in time and space can be gained from records of streamflow and suspended sediment concentration [15,16]. Records of river flow and suspended sediment concentrations at various temporal and spatial scales are crucial because they show patterns in the effects of both natural and human-caused changes to watersheds [17]. However, developing countries such as Ethiopia have few records of the concentration of suspended sediment data.

The majority of Ethiopian rivers lack sediment monitoring stations, and even those that do have them have very scanty and outdated stations. This is due to the expensive and time-consuming laboratory analysis that follows the physical sample collection of suspended sediment concentration [18]. The limited sediment concentration record makes it difficult to plan watershed management, design dams and plan reservoir operations. For water resource planning and watershed management modeling, machine learning and remote sensing techniques can be employed to estimate yearly sediment yield and locate erosion hotspot areas. Therefore, the main goal of this work was to use hydrological and sediment modeling in the Andassa watershed to estimate sediment yield from the watershed and locate erosion hotspot areas with available inadequate data. This study is intended to support engineers, planners, and professionals involved in the design, planning, and management of water resources and watershed development works.

## 2. Materials and Methods

### 2.1. Study Area

Andassa watershed is part of the headstreams of the Upper Blue Nile River. It is situated south of Lake Tana between latitudes of 11°08′ and 11°32′ N and longitudes of 37°16′ and 37°32′ E, respectively (Figure 1). Three administrative districts of the Amhara region—Bahir Dar Zuria, Mecha, and Adet—share 600.6 km$^2$ of the Andassa watershed. Andasa is a perennial river that flows to the main Upper Blue Nile River all year round. The topography of Andassa watershed consists of very rugged hills and valleys, with the lowest point being 1710 m and the highest point being 3216 m a.s.l. (Figure 1a). Over half (58%) of the watershed is composed primarily of flat and gently sloping terrain. The remaining portion of the watershed is primarily hilly and steep. The watershed is characterized by a humid and sub-humid agroclimatic zone. The mean annual rainfall of the watershed during 1990–2020 periods was 1341 mm. The most dominant land use in the watershed (Figure 3) was annual cropland (57%). In comparison to other watersheds in

the Upper Blue Nile Basin, the Andasa watershed had a comparatively high land cover of forest (24%) and shrub (16%). The common soil types of the watershed include Vertisols, Leptosols, and Luvisols which are characterized by clay and clay loam textures (Figure 3). Basalts from the volcanic center, where Termaber basalts and Ashangi basalts are the prominent geological formations, make up the geology of the Andasa watershed.

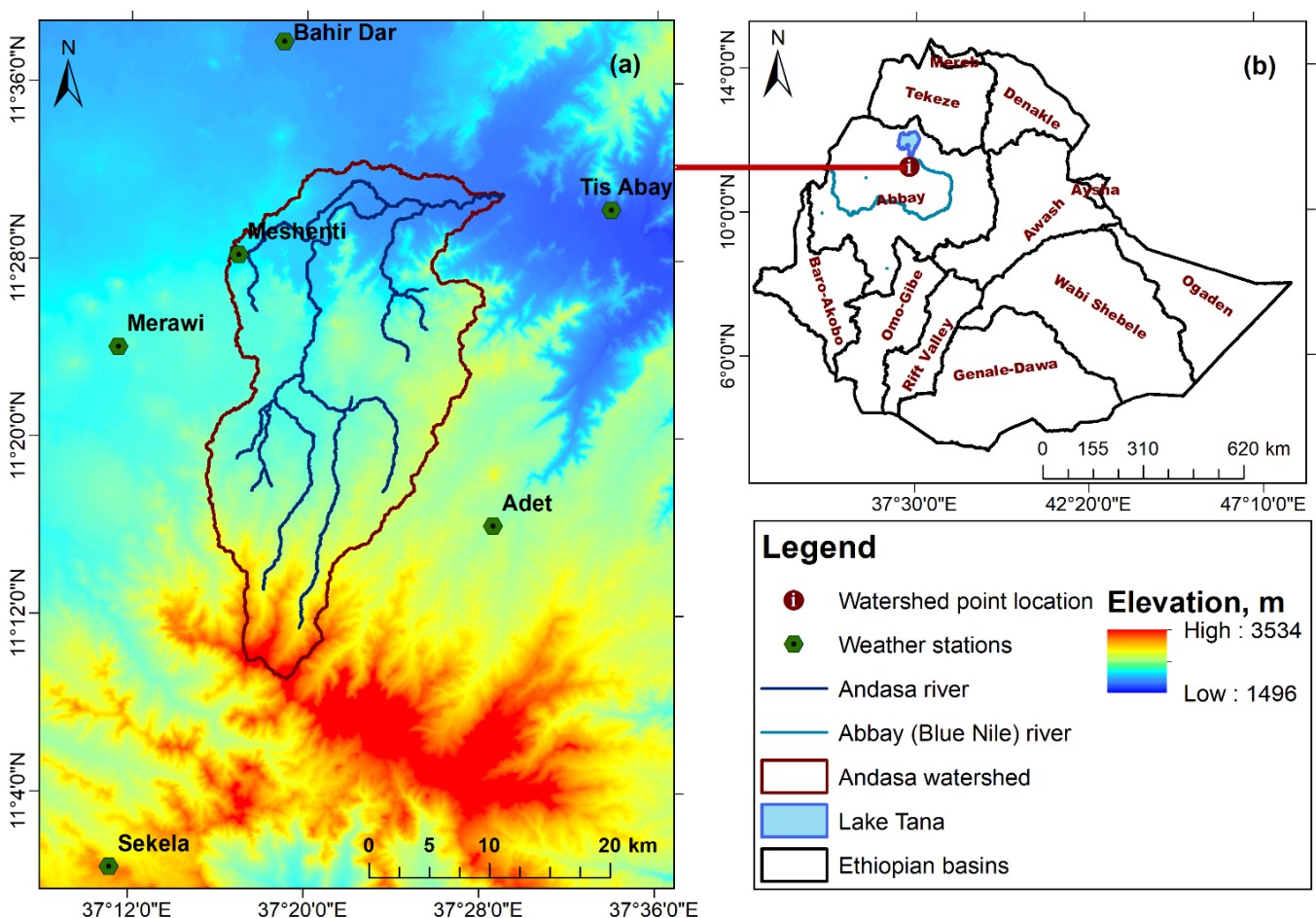

**Figure 1.** Location map of the study area (**a**) Andasa watershed with the location of weather stations and (**b**) Ethiopian basins and the point location of Andasa watershed. The background in the Andasa watershed is the elevation in meters.

*2.2. Dataset*

2.2.1. Time Series Data

This study used daily meteorological data (1990–2012) from Bahir Dar, Merawi, Meshenti, Adet, Tis Abay, and Sekela stations (Figure 1), which were collected from the National Meteorological Agency (NMSA). The five SWAT required meteorological variables such as rainfall, temperature, wind speed, relative humidity, and solar radiation were recorded in Bahir Dar and Adet stations. The Merawi station contain rainfall and temperature while the remaining stations record only rainfall data. Daily observed streamflow for the Andasa River gauging station was acquired from the Ethiopian Ministry of Water and Energy (MoWE) for the period 1990 to 2012. For sensitivity analysis, calibration and validation the daily streamflow was aggregated to monthly flow. Data on suspended sediment concentration of the Andasa watershed were gathered from MoWE at the same time as the streamflow (see the data in Table S1 from the supplementary materials). A sediment yield rating curve was created from the few records of sediment concentration data in order to generate daily sediment yield from the streamflow data. The power equation was fitted to the logarithm of discharge and sediment yield (Figure 2). The daily sediment yield data

from the rating curve was aggregated to monthly for sensitivity analysis, calibration and validation of the SWAT model.

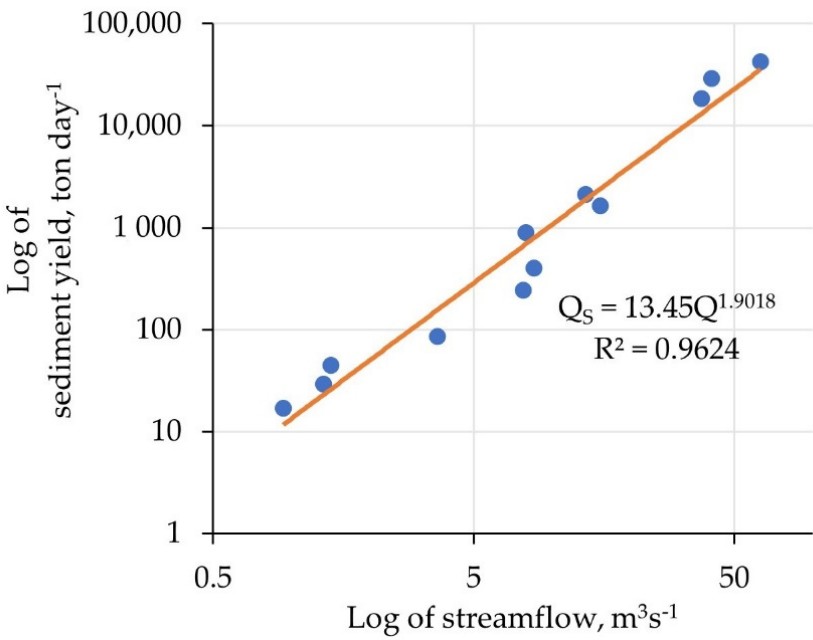

**Figure 2.** The discharge–sediment rating curve of Andasa River prepared for this study.

2.2.2. Spatial Data

The 12.5 × 12.5 m resolution ALOS PALSAR Digital Elevation Model (DEM) was downloaded from the Alaska Satellite Facility for this study. The Advanced Land Observing Satellite-1 (ALOS) PALSAR is the L-band Synthetic Aperture Radar (SAR) which is able to capture the image in all weathers and has day and night observation [19]. The DEM served as an input for the hydrological model to characterize the watershed, sub-watersheds, and drainage patterns using geometric parameters including slope, stream length, etc. The land use/land cover (LULC) for this study was obtained from the Ethiopian Geospatial Information Center (EGIC) with a 30 × 30 m resolution created in 2016 (Figure 3a). The soil map and its associated database, with a resolution of 250 × 250 m, were acquired from African Soil Information Service (AfSIS) in 2014 (Figure 3b). For SWAT modeling, usability of AfSIS soil database performed well for the watersheds of the Upper Blue Nile basin [20]. The hydrological response units (HRUs) were mapped and characterized using slope, LULC, and soil data.

*2.3. Analysis*

2.3.1. Description of SWAT Model

The study employed the Soil and Water Assessment Tool (SWAT) model, which is useful for simulating major hydrological process, effective at predicting long-term impacts, and performed well in predicting streamflow and sediment yield in the watersheds of Upper Blue Nile basin. To predict the runoff, sediment, chemical yields, and crop growth and yield with varying land use, soil, topography, and land management methods over long periods of time on a daily basis, the model may be used for both gauged and un-gauged watersheds [21–23]. The watershed is discretized into a number of sub-watersheds and HRUs in the model, which enables users to simulate with a high degree of spatial detail.

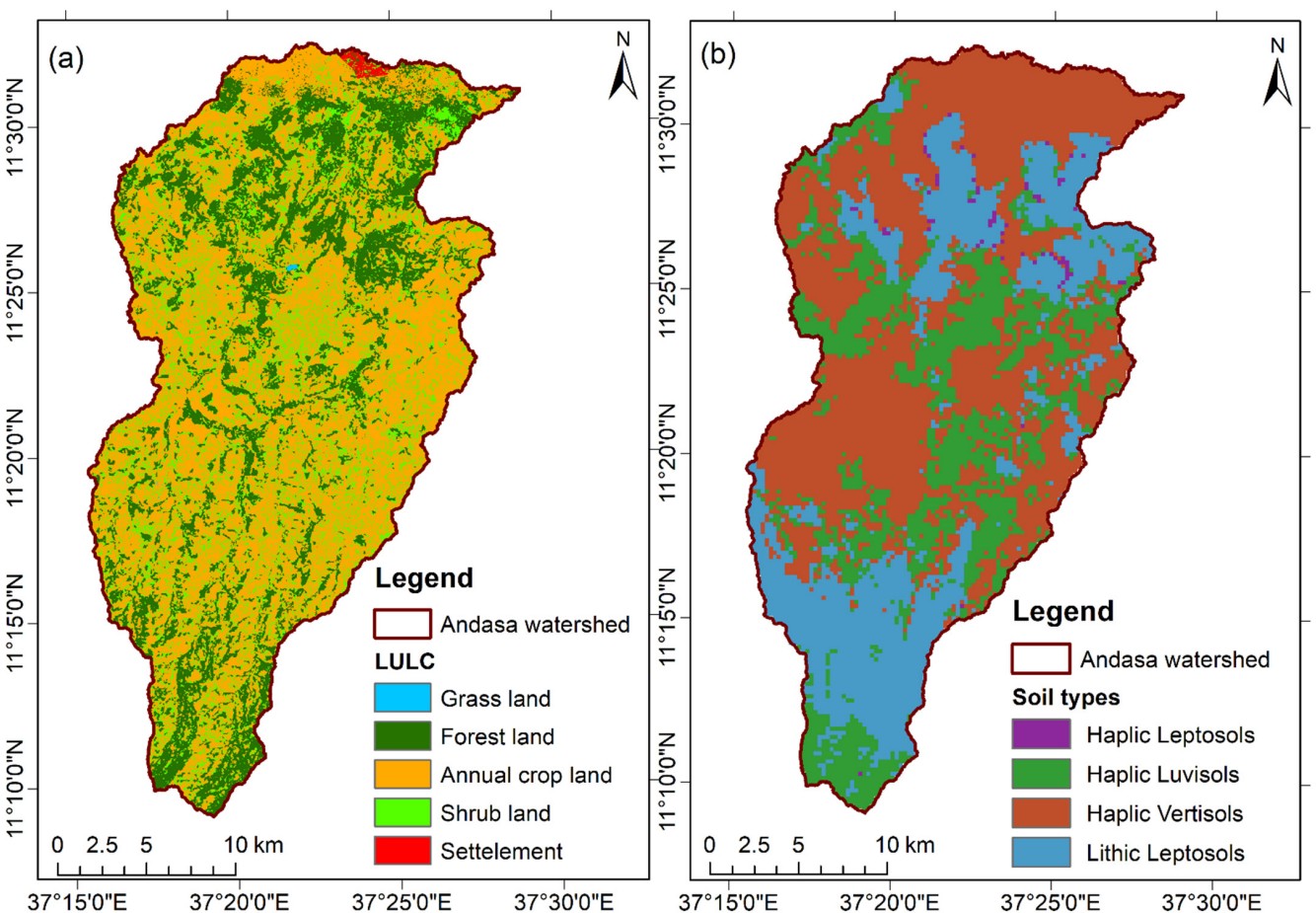

**Figure 3.** Maps of (**a**) land use/land cover and (**b**) soil types of Andasa watersheds.

### 2.3.2. Hydrological Modeling in SWAT

In SWAT, the land phase of a hydrologic cycle of a sub-basin is simulated based on the following water balance equation [24,25].

$$SW_t = SW_o + \sum_{i=1}^{t}(R_{day} - Q_{surf} - E_a - W_{seep} - Q_{gw}) \tag{1}$$

where $SW_t$ is the final soil water content (mm), $SW_o$ is the initial soil water content on the day $i$ (mm), $t$ is the time (days), $R_{day}$ is the amount of precipitation on the day $i$ (mm), $Q_{surf}$ is the amount of surface runoff on the day $i$ (mm), $Ea$ is the amount of evapotranspiration on the day $i$ (mm), $w_{seep}$ is the amount of water entering the vadose zone from the soil profile on the day $i$ (mm), and $Q_{gw}$ is the amount of return flow on the day $i$ (mm). Refer to Neitsch et al. [22] for detailed information and driving equations for each of the water balance components.

SWAT calculates the surface runoff from daily rainfall by using the modified SCS curve number method [25] as indicated in Equation (2) below.

$$Q_{surf} = \frac{\left(R_{day} - 0.2S\right)^2}{R_{day} + 0.8S} \tag{2}$$

where $S$ is the relation parameter in mm and given as

$$S = 25.4\frac{1000}{CN} - 10 \tag{3}$$

where *CN* is the curve number.

SWAT routing phase defines the movement of water, nutrients, sediment and pesticides through the channel network of the watershed into the outlet. In this research, flow was routed through stream network of the watershed from upland areas to the main channel by variable storage routing [26]. Continuity equation was the concept behind storage routing.

For a given reach,

$$\Delta V_{stored} = V_{in} - V_{out} \tag{4}$$

where $V_{in}$ is the volume of inflow during the time step (m$^3$ H$_2$O), and $V_{out}$ is the volume of outflow during the time step (m$^3$ H$_2$O), $\Delta V_{stored}$ is the change in volume of storage during the time step (m$^3$ H$_2$O). The calculation can be further specified as in the Equation (5).

$$V_{stored,2} - V_{stored,1} = \frac{\Delta t}{2}[(q_{in,1} + q_{in,2}) - (q_{out,1} + q_{out,2})] \tag{5}$$

where $q_{in,1}$ is the inflow rate at the beginning of time step in m$^3$ s$^{-1}$, $q_{in,2}$ is inflow rate at the end of time step in m$^3$ s$^{-1}$, $q_{out,1}$ is the outflow rate at the beginning of time step in m$^3$ s$^{-1}$, $q_{out,2}$ is the outflow rate at the end of time step in m$^3$ s$^{-1}$, $\Delta t$ is the length of the time step in second, $V_{stored,2}$ is the storage volume at the end of time step in m$^3$ H$_2$O, and $V_{stored,1}$ is the storage volume at the beginning of time step in m$^3$ H$_2$O.

The volume of water in the channel was divided by the outflow rate to compute the travel time (*TT*).

$$TT = \frac{V_{stored}}{q_{out}} = \frac{V_{stored,1}}{q_{out,1}} = \frac{V_{stored,2}}{q_{out,2}} \tag{6}$$

### 2.3.3. Sediment Modeling in SWAT

SWAT calculates the sediment rate from each HRU by using the Modified Universal Soil Loss Equation using the following equation [27].

$$Sed = 11.8 * \left( Q_{surf} * q_{peak} * Area_{hru} \right)^{0.56} * K_{USLE} * C_{USLE} * P_{USLE} * LS_{USLE} * CFRG \tag{7}$$

where *Sed* is the yield of sediment (ton day$^{-1}$), $Q_{surf}$ is the volume of surface runoff (mm ha$^{-1}$), $q_{peak}$ is the peak surface runoff rate (m$^{-3}$s$^{-1}$), $Area_{hru}$ is hydrologic response unit area (ha), $K_{USLE}$ is USLE soil erodibility factor, $C_{USLE}$ is USLE cover factor, $LS_{USLE}$ is USLE topography factor and $P_{USLE}$ is USLE soil protection factor, and *CFRG* accounts for stoniness.

For channel networks, SWAT computes the sediment flow as [22]:

$$Sed_{ch} = Sed_{chi} - Sed_{dep} + Sed_{deg} \tag{8}$$

where $Sed_{ch}$ is the amount of suspended sediment in the reach, $Sed_{deg}$ is the amount of sediment that reenters the reach segment, $Sed_{dep}$ is the amount of sediment deposited in the reach segment, and $Sed_{chi}$ is the amount of suspended sediment in the reach at the beginning of the time period. The unit of all the parameters in the equation is metric tons.

Similarly, SWAT calculates the amount of sediment transported out of reach as [22]:

$$Sed_{out} = Sed_{ch} * \left( \frac{V_{out}}{V_{ch}} \right) \tag{9}$$

where $Sed_{out}$ is the amount of sediment transported out of the reach in tons, $Sed_{ch}$ is the amount of suspended sediment in the reach in tons, $V_{ch}$ is the volume of water in the reach segment in m$^3$, and $V_{out}$ is the volume of outflow during the time step in m$^3$.

2.3.4. SWAT Model Setup, Sensitivity Analysis, Calibration, and Validation

The time series and geographical input data were first created in accordance with the SWAT model's specifications. During data preparation, ArcGIS and Microsoft Excel were also used. Watershed and stream network definition (Figure 4a), HRU analysis (Figure 4b), and model simulation (Figure 4c) are important steps in SWAT modeling. The SWAT model output is used for sensitivity analysis (Figure 4d), calibration and validation (Figure 4e) with the monthly streamflow and sediment yield data using the SWAT Calibration and Uncertainty Procedures (SWAT-CUP). The discretization for the Andasa watershed is created in SWAT with an area threshold of 1.3 km$^2$. The watershed discretization provided 21 sub-watersheds for the entire Andasa watershed. The HRU Analysis takes land use, soil, and slope, to divide each sub-basin into HRUs, with specific combinations of the three layers.

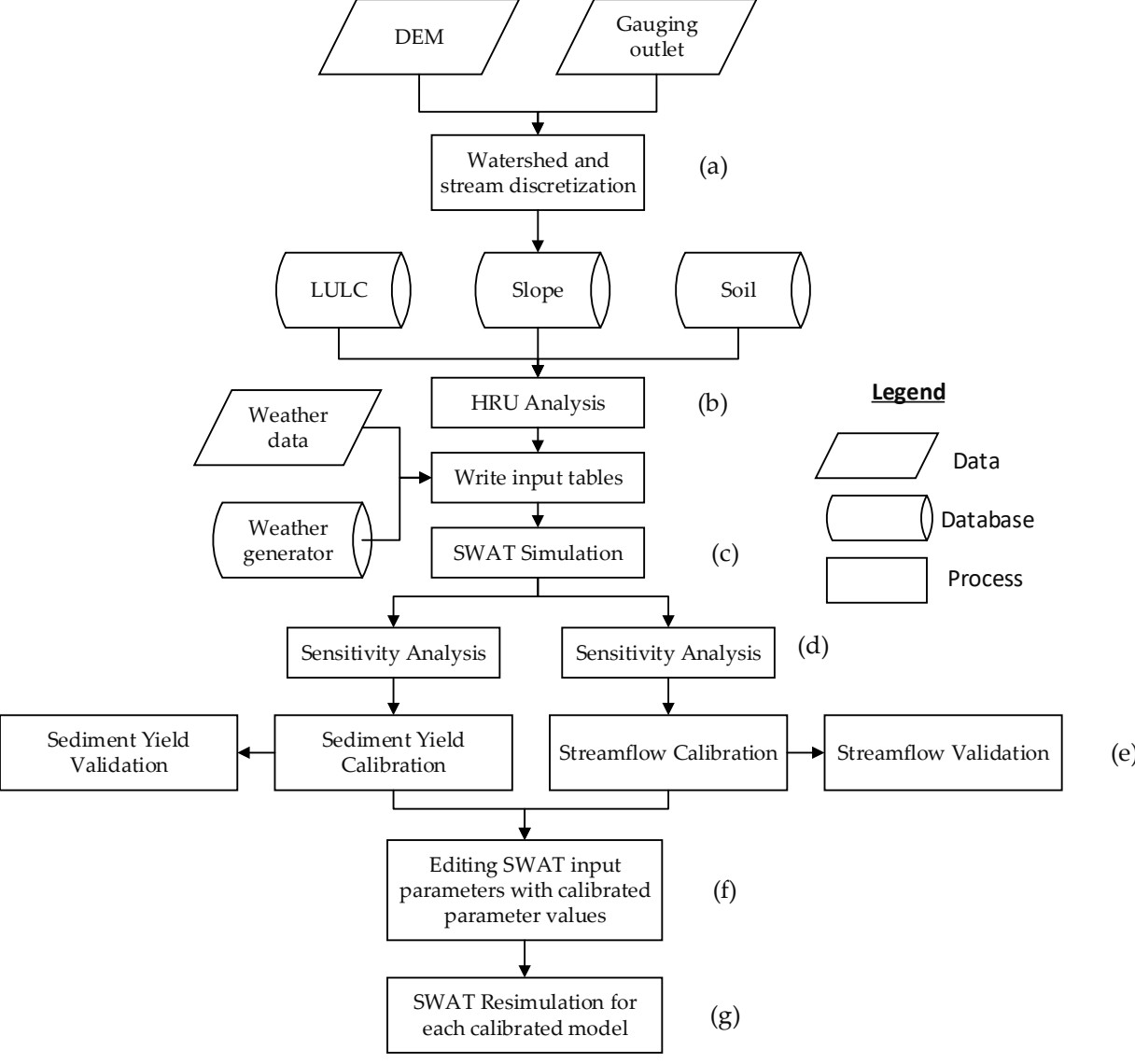

**Figure 4.** Workflow chart showing hydrological and sediment modeling using SWAT (modified from Adem et al. [20]): (**a**) watershed delineation and stream discretization using DEM and gauging outlet, (**b**) HRU analysis using LULC, soil and slope, (**c**) SWAT simulation after writing input tables, (**d**) identifying sensitive parameters for streamflow and sediment yield, (**e**) streamflow and sediment yield calibration and validation, (**f**) editing initial model parameters with calibrated parameters, and (**g**) model restimulation for output discretization.

The layer produced by this process is crucial to the ultimate analysis performed by the SWAT model. HRUs determine how land will respond to rainfall, runoff, infiltration, and other hydrologic processes during the simulation. Each sub-basin can then have one or more major HRUs within it. HRUs are created by a 10% threshold for slope, soil, and LULC definition [28]. Accordingly, a total of 406 HRUs are created for the entire watershed (Figure 4b). After the writing the input tables (terrain, land use, soil and weather), the model simulates streamflow and sediment yield using the equations indicated in Sections 2.3.2 and 2.3.3 (Figure 4c). In this study, the Penman–Monteith method of evapotranspiration estimation was used.

The SWAT model is calibrated using the Sequential Uncertainty Fitting Version-2 (SUFI-2) optimization algorithm included inside the SWAT-CUP. The SUFI-2 algorithm is selected for its satisfactory performance in the Upper Blue Nile Basin [29]. SWAT model users frequently face difficulties in obtaining influential parameters for calibration. In such cases, sensitivity analysis is helpful to identify and rank parameters that have a substantial effect on specific model outputs of interest. Since certain parameters in this study are sensitive to both flow and sediment, some are just sensitive to sediment, and others are only sensitive to flow, the sensitivity analysis for each was conducted independently [30]. Streamflow and sediment yield data from the years 1992 to 2005 are used for calibration of the Andasa SWAT model, which was then used for validation in the years 2006 to 2012. The streamflow and sediment yield data for the period 1990 to 1991 are used for model warm-up. The best-fitted parameters during the calibration process were used to edit the initial model parameters in the Andasa SWAT model (Figure 4f). The model was re-simulated to obtain streamflow and sediment predictions from the entire watershed and each sub-watersheds (Figure 4g).

### 2.3.5. Identification of Erosion Hotspot Areas

Erosion hotspot areas were identified from the calibrated model output to prioritize the areas for soil and water conservation (SWC) practices. The mean annual sediment yield of the sub-watersheds is classified using the soil erosion severity class developed by Tilahun et al. [31]. As a result, the soil erosion level in the Andasa watershed is classified into four classes, namely, negligible erosion class ($\leq$3 ton ha$^{-1}$ yr$^{-1}$), acceptable erosion class (3 to $\leq$4.5 ton ha$^{-1}$ yr$^{-1}$), moderately eroded class (4.5 to $\leq$9.5 ton ha$^{-1}$ yr$^{-1}$), and severely eroded (>9.5 ton ha$^{-1}$ yr$^{-1}$) class categories [31].

### 2.3.6. Model Performance Evaluation and Statistics

It is essential to assess the predicting model's performance in terms of accuracy, consistency, and adaptability when making hydrological and sediment predictions [32]. Using statistical criteria and visual examination, the streamflow and sediment yield prediction capabilities of the SWAT model were assessed. In order to quantify how well the predicted results match with the observed data during a certain time period, descriptive statistics and performance measures including the coefficient of determination ($R^2$), Nash–Sutcliffe Efficiency (NSE), and percent of bias (PBIAS) were utilized. Performance criteria used in this study are similar with Adem et al. [20]. The Mann–Kendall trend test was used to determine whether or not there is a significant trend ($p < 0.5$) in rainfall, evapotranspiration, streamflow, and sediment yield in the simulation period (1992–2020).

## 3. Results

### 3.1. Streamflow Sensitivity Analysis, Calibration, and Validation

During the study period, 23 streamflow parameters (Table S2 from the supplementary material) were tested for sensitivity, and the result revealed that 14 parameters are more sensitive (Table 1). When compared to the selected parameters, the effect of the remaining nine parameters on the model output was minor. The first four highly sensitive parameters for the streamflow were the SCS runoff curve number (CN_2), soil available water content (SOL_AWC), depth from the soil surface to the bottom layer (SOL_Z), and threshold depth

of water in the shallow aquifer required for return flow to occur (GWQMN). Streamflow calibration was conducted using the selected 14 sensitive parameters tabulated in Table 1. The fitted parameter values that obtained from the calibration process at the same model inputs can be used for future watershed applications.

**Table 1.** Sensitive parameters for observed streamflow, initial range, and fitted value. In the parameters list, r_ stands for the operation of the existing parameter value multiplied by (1+ given value), v_ is for the operation of the existing parameter value to be replaced by a given value, and a_ is for the operations of a given value added to the existing parameter value.

| Parameters with Operation | Description | Fitted Values (Sensitivity Rank) | Parameter Initial Range |
|---|---|---|---|
| r_CN2 | SCS runoff curve number | 0.042 (1) | −0.2–0.2 |
| r_SOL_AWC | Available water capacity of the soil layer, mm $H_2O$/mm soil | −0.006 (2) | −0.2–0.2 |
| r_SOL_Z | Depth from the soil surface to the bottom of the layer, mm | −0.156 (3) | −0.2–0.2 |
| a_GWQMN | Threshold depth of water in the shallow aquifer required for return flow to occur, mm $H_2O$ | −588 (4) | −1000–1000 |
| v_RCHRG_DP | Deep aquifer percolation fraction | 0.565 (5) | 0–1 |
| a_GW_DELAY | Groundwater delay, days | 14.8 (6) | −30–60 |
| v_GW_REVAP | Groundwater "revap" coefficient | 0.146 (7) | −0.036–0.2 |
| v_ALPHA_BF_D | Base flow alpha factor for groundwater recession of the deep aquifer, 1/days | 0.144 (8) | 0–1 |
| a_CANMX | Maximum canopy storage, mm $H_2O$ | 0.112 (9) | 0–10 |
| v_CH_K2 | Effective hydraulic conductivity in the main channel alluvium, mm/h | 12.15 (10) | 0–15 |
| v_ALPHA_BF | Base flow alfa factor, days | 0.46 (11) | 0–1 |
| v_SURLAG | Surface runoff lag time, days | 3.313 (12) | 0–10 |
| v_BIOMIX | Biological mixing efficiency | 0.433 (13) | 0–10 |
| v_ESCO | Soil evaporation compensation factor | 0.991 (14) | 0–1 |

The SWAT model performed well in both the calibration and validation periods in the Andasa watershed, with coefficient of determination ($R^2$) of 0.88, Nash–Sutcliffe efficiency (NSE) of 0.88, percent of bias (PBIAS) of 6.4 (Figure 5a) and $R^2$ of 0.81, NSE of 0.80, PBIAS of 9.9, respectively Figure 5b). Positive values of PBIAS indicate that the model underpredicts the observed streamflow. During the calibration and validation period, the slope of the trend line in the scatter plot of observed versus the simulated was between 0.9 and 0.82 (Figure 5). The calibration intercept was less than 0.5 and the validation intercept was greater than 0.5 (Figure 5). The simulated streamflow hydrograph also best described the observed streamflow during the calibration period (Figure 6a). The SWAT model best predicted observed flow in the majority of the hydrograph's low flow, rising, and falling limbs (Figure 6a). The model underpredicted the peaks of the observed streamflow. The model responds to multiple rainfall peaks than the observed flow in a single year (see 1997, 1998 and 2006 in Figure 6a). Despite satisfactory model performance during the validation period, the model underpredicted the low flows and overpredicted the peak flows (Figure 6b). There is no well-defined prediction for the rising and falling limb of the hydrographs, with under-predicting in some years and over-predicting in other years (Figure 6b).

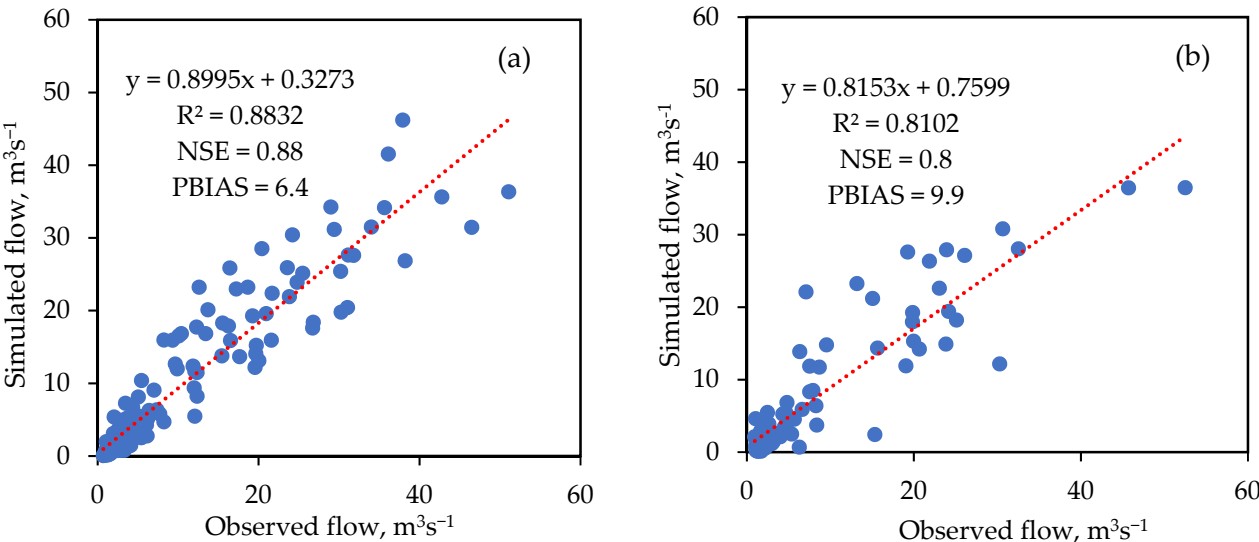

**Figure 5.** Scatter plot of observed versus simulated streamflow during (**a**) calibration and (**b**) validation periods. $R^2$ is the coefficient of determination; NSE is the Nash–Sutcliffe efficiency, and PBIAS is the percent of bias.

**Figure 6.** Observed and simulated streamflow with rainfall during (**a**) calibration and (**b**) validation periods.

### 3.2. Water Balance Components before and after Calibration

The mean annual water balance components differed significantly before and after calibration, owing to the calibration process's goal of minimizing the difference between the predicted and observed streamflow (Table 2). After calibration, the simulated surface runoff was 21% higher than before calibration. This is due to a decrease in evapotranspiration, baseflow (return flow), and percolation following model calibration. The decrease in baseflow is a direct result in percolation to the shallow aquifer. The model calibration process significantly reduced the shallow aquifer evaporation from 46 to 0.2 mm (Table 2). After calibration, approximately 80% (37 mm) of the evaporation from the shallow aquifer was retained as recharge. In other words, recharge to the deep aquifer was increased from 5 mm (before calibration) to 42 mm after calibration (Table 2). Before calibration, the water balance shows that evapotranspiration accounts for 72% of the mean annual rainfall, surface runoff accounts for 19% of the mean annual rainfall; lateral flow accounts for 8.7% of the surface runoff, and baseflow flow accounts for 21% of the surface runoff. The ratio of total streamflow (surface runoff, lateral and baseflow), to rainfall in the Andasa watershed was 0.25 prior to the model calibration (Table 2). This means that 25% of the rainfall was flowing out from the watershed in the form of streamflow. After calibration, average surface runoff, streamflow water yield, percolation from the soil layer, and evapotranspiration account for 24.2%, 31.3%, 5.5%, and 68.9% of the watershed's mean annual rainfall, respectively. Baseflow accounts for 10% of the mean annual surface runoff, while the mean annual soil water storage accounts for 12% of the mean annual rainfall.

**Table 2.** Mean annual water balance component before and after calibration in millimeter depth.

| Hydrological Component | Before | After |
|---|---|---|
| Rainfall | 1341.2 | 1341.2 |
| Evapotranspiration | 964.4 | 924.7 |
| Surface Runoff | 256.73 | 324.65 |
| Lateral flow | 22.35 | 19.96 |
| Percolation to the shallow aquifer | 99.86 | 74.63 |
| Return flow | 53.71 | 32.49 |
| Recharge to the deep aquifer | 4.99 | 42.15 |
| Revap from the shallow aquifer | 45.57 | 0.02 |

### 3.3. Sediment Yield Sensitivity Analysis, Calibration, and Validation

To assess the quantity of watershed sediment yield, initial sediment parameters were examined and ranked by the global sensitivity analysis procedure using the sediment yield data (Table 3). The sensitivity analysis result revealed that the USLE cover factor (USLE_C), USLE soil erodibility factor (USLE_K), and Channel erodibility factor (CH_COV1) are the first three and most sensitive parameters for the sediment yield data generated using sediment rating curve. The other parameters also impact the sediment yield prediction and are utilized for calibration; see the detail sensitivity rank in Table 3. All the parameters that potentially influence sediment prediction were used for calibration in this investigation.

SWAT model predicted sediment yield with $R^2$ of 0.79, NSE of 0.72, and PBIAS of 2.7 during the calibration and with $R^2$ of 0.7, NSE of 0.66, and PBIAS of $-8.6$ during the validation periods (Figure 7). In contrast to the streamflow, the PBIAS indicated that the model over-predicted the sediment yield generated from the sediment yield rating curve over the validation period. In comparison to the stream flow calibration, the PBIAS during the calibration was minimal. The trend equation's lower slope and the higher intercept indicated a bias in the predicted sediment yield compared with the predicted streamflow (Figures 5 and 7). The hydrograph of the simulated sediment yields, similarly to the streamflow prediction, accurately described the sediment yield from the rating curve throughout the calibration period (Figure 8a). In low flow times, there was extremely high agreement between the predicted and observed rating curve sediment yield. Except for the peaks of 1992, 1998, 2000, and 2003, the model over-predicted the sediment yield in

the sediment hydrograph peaks, falling limbs, and rising limbs (Figure 8a). The model, similarly to the streamflow prediction, responds to multiple rainfall peaks (zigzag line in the peaks of Figure 8a,b). Despite acceptable model performance over the validation period, the model overpredicted the sediment yield with the exception of the 2006, 2010, and 2011 peaks (Figure 8b).

**Table 3.** Sensitive parameters for sediment yield, initial range, and fitted values. In the parameters list, r_ stands for the operation of the existing parameter value multiplied by (1+ given value); and v_ is for the operation of the existing parameter value to be replaced by a given value.

| Parameters with Operation | Description | Fitted Values (Sensitivity Rank) | Parameter Initial Range |
|---|---|---|---|
| v_USLE_C | The minimum value of the USLE C factor for land cover/plant | 0.183 (1) | 0.001 to 0.5 |
| v_USLE_K | USLE soil erodibility (K) factor, ton/m$^2$ h | 0.510 (2) | 0 to 0.65 |
| v_CH_COV1 | Channel erodibility factor | 0.225 (3) | 0 to 1 |
| r_SPCON | Linear parameter for calculating the maximum amount of sediment that can be re-entrained during channel sediment routing | 0.0092 (4) | 0.008 to 0.01 |
| v_ADJ_PKR | Peak rate adjustment factor for sediment routing in the sub-basin (tributary channels) | 1.768 (5) | 0 to 2 |
| v_USLE_P | USLE support practice factor | 0.295 (6) | 0 to 1 |
| v_CH_COV2 | Channel cover factor | 0.705 (7) | 0 to 0.6 |
| v_SPEXP | Exponent parameter for calculating sediment re-entrained in channel sediment routing | 1.028 (9) | 1 to 1.5 |

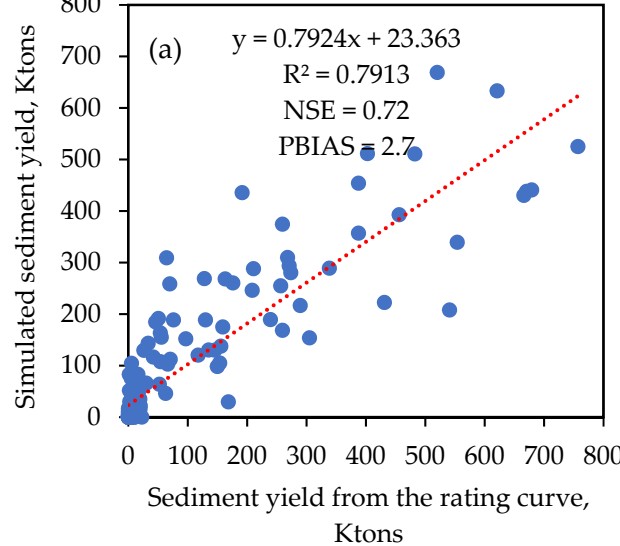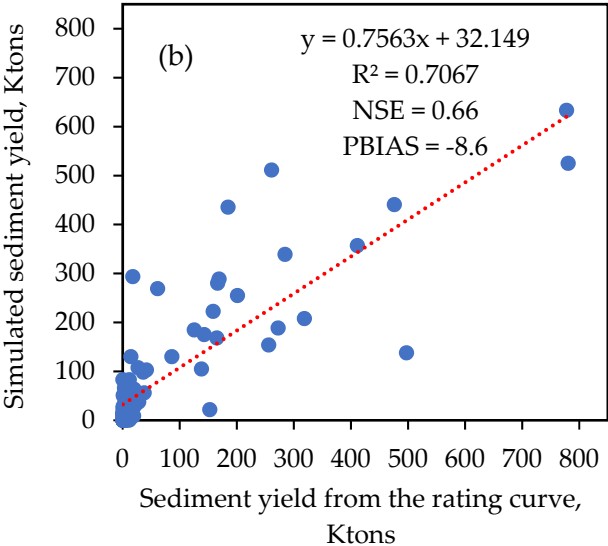

**Figure 7.** Scatter plot of sediment yield from the rating curve versus simulated streamflow during (**a**) calibration and (**b**) validation periods. $R^2$ is the coefficient of determination, NSE is the Nash–Sutcliffe efficiency, PBIAS is the percent of bias, and Ktones stands for thousands of tons.

### 3.4. Temporal Variability of Sediment Yield in the Andasa Watershed

The SWAT model estimate, the mean monthly sediment yield between 1992 and 2020, revealed that the sediment yield was higher in the months with more rainfall (Figure 9). In June, July, August, and September, the total annual sediment yield was about 80% (Figure 9). Only 20% of the sediment yield was contributed by the remaining months of the year. Although the rainfall and streamflow peaked in July, August was the month with the highest sediment yield. In accordance with the 368.7 mm of mean monthly precipitation that fell in July, 104.7 mm of mean monthly streamflow and 4.9 t ha$^{-1}$ month$^{-1}$ of mean monthly sediment yield were recorded. In August, the average monthly yield of sediment was 5.52 t ha$^{-1}$ month$^{-1}$, but the corresponding rainfall and streamflow were 288 mm and

102 mm, respectively. Concentrated streamflow was observed at the beginning of the rainy season, similar to other watersheds of the Ethiopian highlands.

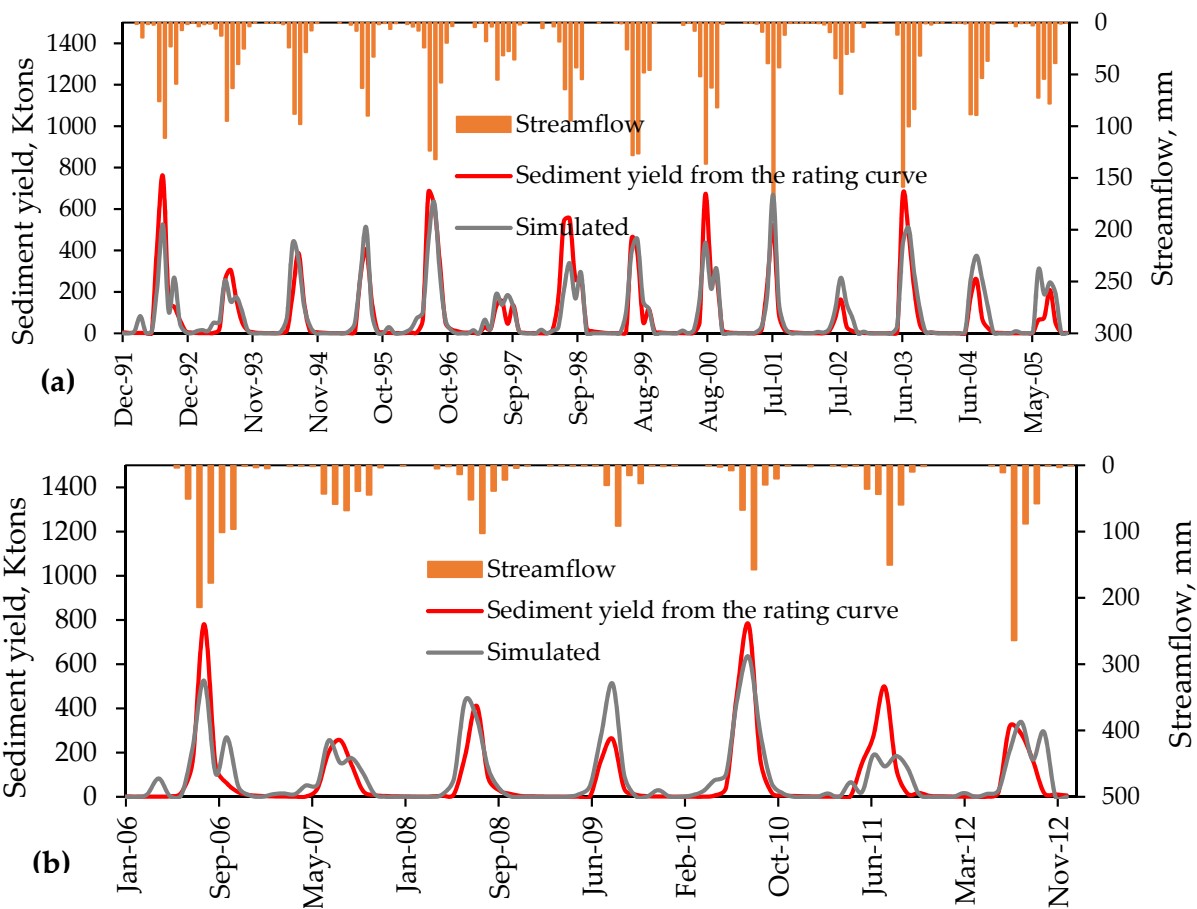

**Figure 8.** Observed and simulated sediment yield with runoff during (**a**) calibration and (**b**) validation and periods. Ktones stands for thousands of tons.

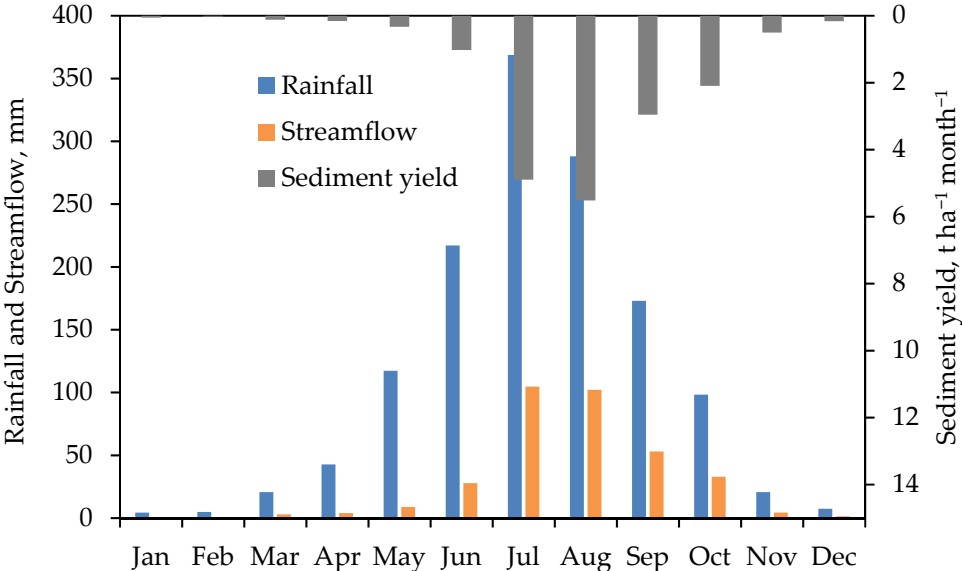

**Figure 9.** Mean monthly rainfall, streamflow, and sediment yield of the Andasa watershed between 1990 and 2020.

Following model calibration, the average annual sediment yield (1992–2020) at the watershed outlet ranged from 5.9 to 50.6 t ha$^{-1}$yr$^{-1}$ with an average of 17.9 t ha$^{-1}$yr$^{-1}$ (Figure 10). Rainfall and streamflow directly influenced the Andasa watershed's annual sediment yield. For instance, the highest streamflow (721 mm) and annual rainfall (1852 mm) occurred in the year 2006, resulting in an average sediment yield of 50.6 t ha$^{-1}$yr$^{-1}$, the highest yield during the study period (1992–2020). On the other hand, the year 2009 had the lowest annual rainfall (1079.6 mm), the smallest relative streamflow (216 mm), and the highest annual sediment yield (9.6 t ha$^{-1}$yr$^{-1}$). This is directly connected to the fact that sediment yield and streamflow and rainfall have a positive relationship (Figure 11). However, compared to the correlation between the rainfall and sediment yield ($R^2 = 0.62$), the correlation between streamflow and sediment yield was stronger ($R^2 = 0.98$). However, none of the trends were statistically significant (p 0.5), despite the fact that the annual rainfall of the watershed was increasing and the streamflow and sediment yield were decreasing (Figure 10). The increase in water loss in the form of evapotranspiration was the cause of the declining trend in streamflow and sediment yield (Figure 10).

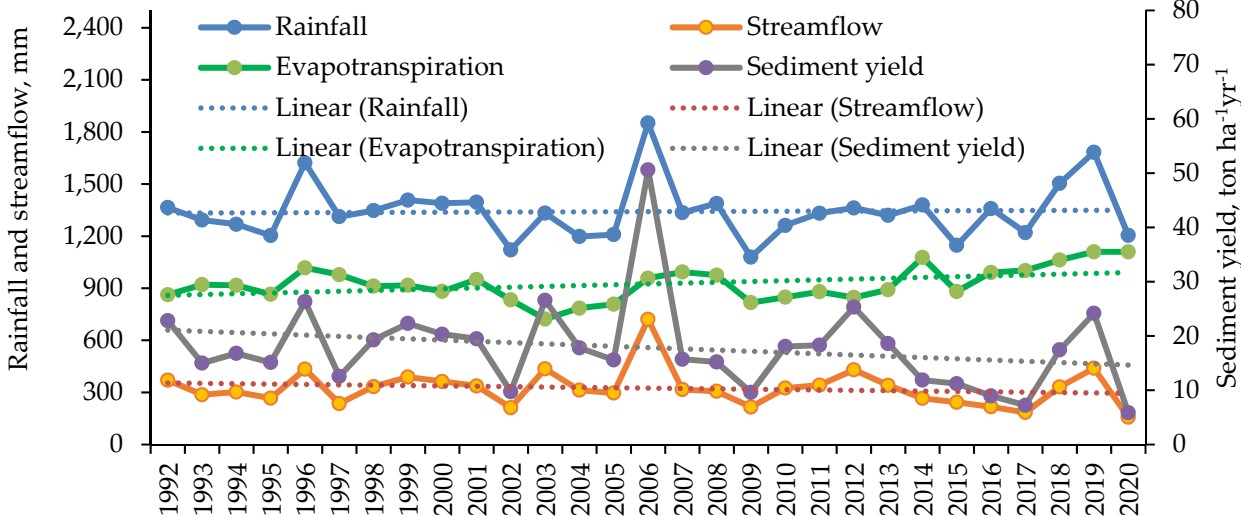

**Figure 10.** Annual rainfall, evapotranspiration, streamflow, sediment yield, and the corresponding trend lines for the entire simulation period.

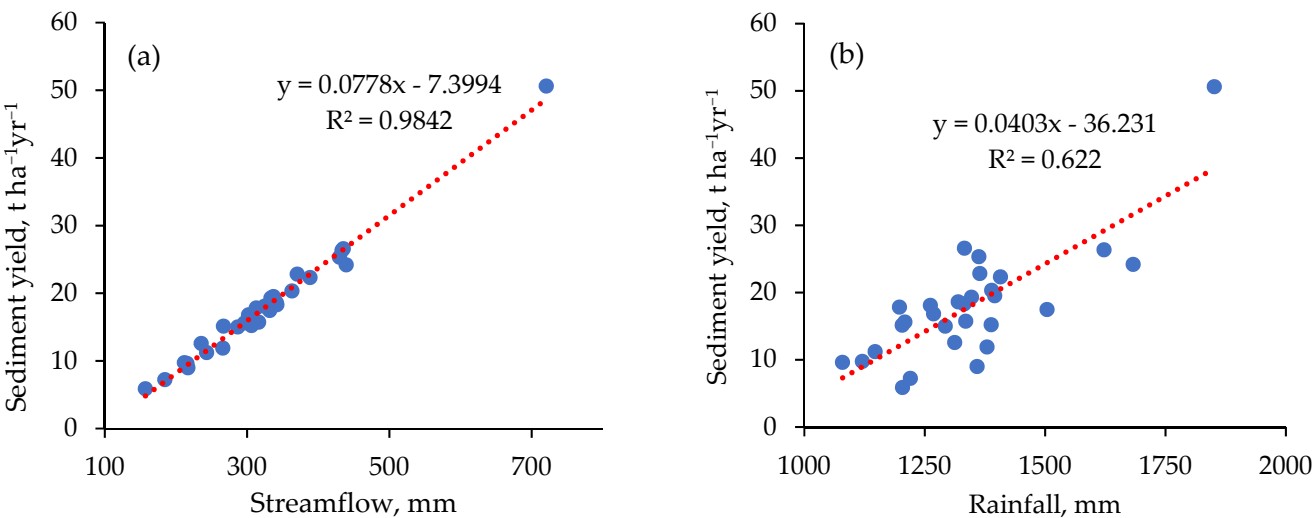

**Figure 11.** Correlation of (**a**) annual streamflow and sediment yield and (**b**) annual rainfall and sediment yield.

*3.5. Spatial Distribution of Sediment Yield in the Andasa Watershed*

The spatial distribution of sediment yields in the Andasa watershed indicated that 4 and 10 of the 21 sub-watersheds were severely and moderately eroded, respectively (Figure 12). The remaining seven sub-watersheds had acceptable and barely negligible rates of erosion. In the seven severely eroded sub-watersheds of Andasa, cultivated land in Luvisols and Vertisols made up 21% of the watershed (Figure 12). On the steep slopes of these watersheds, there were also shrub lands with shallow Leptosol soil depths and degraded soils. The D hydrological soil group, which has high potential for runoff and low rate of infiltration, dominates the four sub-watersheds since the soils of the watershed are characterized by clay and clay loam (Figure 3). The majority of these erosion hotspot areas are found in the middle and lower slopes of the Andasa watershed (Figure 12). The Bahir Dar Zuria district is home to 84% of the sub-watersheds. The Andasa watershed was moderately eroded in more than half (55%) of its area (Figure 12). Even though watershed management strategies for erosion-prone areas should be prioritized, conservation efforts for moderately eroded areas should also be prioritized to prevent further degradation. The central and northwestern regions of the Andasa watershed contain 23% of the watershed's areas that are not at risk for erosion.

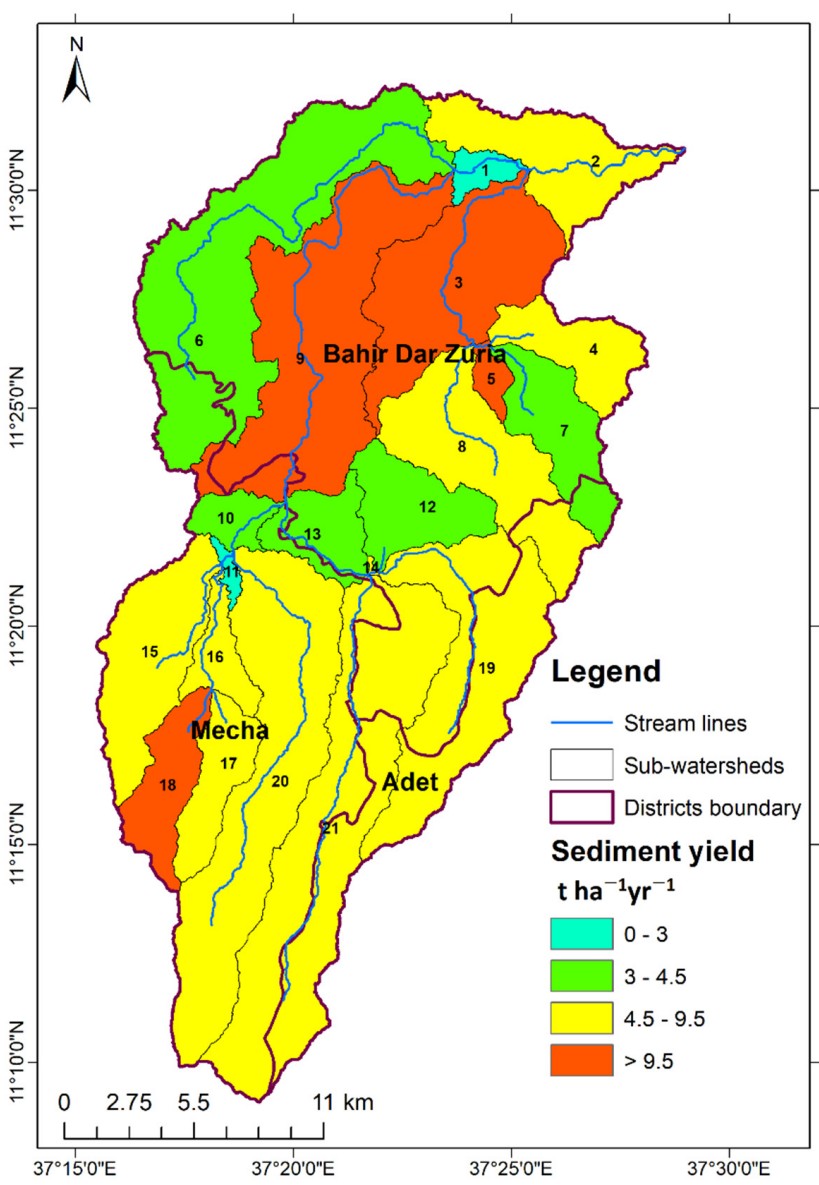

**Figure 12.** Spatial variability of sediment yield in the sub-watersheds of Andasa.

## 4. Discussion

### 4.1. SWAT Model Performance in the Upper Blue Nile Basin

In this study, the SWAT model prediction between 1992 and 2012 revealed that the model did a good job of predicting the hydrology and sediment yield in the Andasa watershed (Figures 5 and 7). The model's prediction performance values agreed with published studies in the Abbay (Upper Blue Nile) basin watersheds (Table 4). Table 4 shows that the model performed well for both small and large watersheds (areas ranging from 1.13 to 174,166 km$^2$). The model performed well in every watershed during all of the calibration and validation phases. Most of the data used in these studies were collected from the Ministry of Water and Energy (MoWE) of Ethiopia. The results did not show the SWAT model performance for recent years because the watershed data had not been updated for more than ten years. The predicted and the observed streamflow and sediment yield of the watersheds were out of sync in the majority of the studies indicated in Table 4. This is attributed to the model's limitations in simulating peaks [33] or inadequate representation of the rainfall with weather stations in the watersheds [34,35]. Therefore, the SWAT model needs enhancements in peak flow predictions, even though it is a promising long-term continuous simulation model in watersheds of the Upper Blue Nile basin [33].

**Table 4.** Performance of SWAT model in predicting watershed hydrology and sediment yield in watersheds of the Abbay (Upper Blue Nile) basin in a monthly time step. The superscript D in some watershed names indicates the daily time step of the modeling.

| Watershed | Area, km$^2$ | Calibration | | | | Validation | | | | Source |
|---|---|---|---|---|---|---|---|---|---|---|
| | | R$^2$ | NSE | PBIAS | Period | R$^2$ | NSE | PBIAS | Period | |
| **Streamflow** | | | | | | | | | | |
| Abbay at Eldiem | 174,166 | 0.85 | 0.83 | −4.7 | 2001–2009 | 0.89 | 0.88 | 8.3 | 2010–2014 | [36] |
| Abbay at Kessie | 64,728 | 0.81 | 0.68 | −10.8 | 2001–2009 | 0.93 | 0.89 | 9.7 | 2010–2015 | [36] |
| Rib | 1316 | 0.83 | 0.78 | 7 | 1996–2007 | 0.7 | 0.41 | 53 | 2008–2013 | [20] |
| Gumara | 1464 | 0.87 | 0.76 | 3.29 | 1998–2002 | 0.83 | 0.68 | −5.4 | 2003–2005 | [37] |
| Awramba | 7 | 0.98 | 0.94 | −16.4 | 2014–2017 | 0.97 | 0.96 | −0.1 | 2017 | [38] |
| Gilgel Abay [D] | 1654 | 0.8 | 0.77 | - | 1996–2004 | 0.76 | 0.75 | - | 2005–2008 | [39] |
| Nashe | 946 | 0.89 | 0.82 | 5.7 | 1987–1999 | 0.88 | 0.85 | 8.6 | 2000–2008 | [40] |
| Gomit | 3.59 | 0.7 | 0.63 | 14 | 2015–2017 | - | - | - | - | [20] |
| Main Beles | 3485 | 0.82 | 0.81 | −8.4 | 1995–2002 | 0.8 | 0.78 | 1.84 | 2003–2010 | [41] |
| Anjeni | 1.13 | 0.9 | 0.89 | - | 1984–1988 | 0.91 | 0.89 | - | 1989–1993 | [42] |
| Koga | 287 | 0.65 | 0.58 | 24.5 | 1992–2001 | 0.67 | 0.58 | 8.8 | 2002–2007 | [43] |
| Minchet | 1.13 | 0.94 | 0.93 | - | 1986–1998 | 0.92 | 0.92 | - | 2010–2014 | [44] |
| Guder | 7011 | 0.75 | 0.73 | −12.9 | 1990–2004 | 0.81 | 0.79 | 11.4 | 2005–2008 | [45] |
| **Sediment yield** | | | | | | | | | | |
| Abbay at Eldiem [D] | 184,560 | - | 0.88 | −0.05 | 1990–1996 | - | 0.83 | −11 | 1998–2003 | [34] |
| Gumara | 1250 | 0.68 | 0.67 | −6.1 | 1995–2002 | 0.7 | 0.69 | −11.2 | 2003–2007 | [28] |
| Gilgel Abbay [D] | 1654 | 0.59 | 0.58 | - | 1996–2004 | 0.56 | 0.51 | - | 2005–2008 | [46] |
| Main Beles | 3485 | 0.81 | 0.8 | 5 | 1995–2002 | 0.79 | 0.75 | 5 | 2003–2010 | [41] |
| Anjeni | 1.13 | 0.85 | 0.81 | 28 | 1984–1988 | 0.8 | 0.79 | 30 | 1989–1993 | [42] |
| Koga | 287 | 0.75 | 0.73 | 7.8 | 1991–2000 | 0.8 | 0.79 | 6.4 | 2002–2007 | [43] |
| Minchet | 1.13 | 0.71 | 0.53 | - | 1986–1998 | 0.86 | 0.84 | - | 2010–2014 | [44] |
| Guder | 7011 | 0.8 | 0.78 | −12.3 | 1991–2004 | 0.84 | 0.81 | 14.24 | 2005–2008 | [45] |

There were few suspended sediment concentration data in Ethiopian rivers, and the majority of the records were taken in the rainy season. This is attributed to the financial and time constraints of the country. The data limitation in time and space challenges engineers, planners, and practitioners in designing, planning, and management of water resource and watershed development works. This study demonstrates how to use the SWAT model to estimate the sediment yield and locate erosion hotspot areas with sparse sediment concentration data. Both globally [47–49] and in Ethiopia [27,40,42,44,45], similar approaches were used. The SWAT model accurately predicted the sediment yield from the

rating curve in all watershed sizes of the Upper Blue Nile basin (Table 5). The model slightly overestimated the mean annual sediment yield of the watersheds with the exception of the Anjeni watershed. This study also confirmed the satisfactory performance of the SWAT model in predicting sediment yield in the Andasa watershed between 1992 and 2012. According to Table 5, the mean annual sediment yield of Andasa watershed ranged from 5 to 28 t ha$^{-1}$ yr$^{-1}$.

**Table 5.** Predicted and observed mean annual sediment yield prediction of SWAT model in watersheds of the Abbay (Upper Blue Nile) basin at a monthly time step. The superscript D in some watershed names indicates the daily time step of the modeling.

| Watershed | Area, km$^2$ | Observed, t ha$^{-1}$ yr$^{-1}$ | Predicted, t ha$^{-1}$ yr$^{-1}$ | Data Type | Period | Source |
|---|---|---|---|---|---|---|
| Abbay at Eldiem [D] | 184,560 | 6.3 | 7.1 | Observed | 1998–2003 | [34] |
| Gumara | 1250 | 19.7 | - | Rating curve | 2003–2007 | [28] |
| Gilgel Abay [D] | 1654 | 19 | 20.8 | Rating curve | 2005–2008 | [46] |
| Main Beles | 3485 | 4.8 | 5.5 | Rating curve | 2003–2010 | [41] |
| Anjeni | 1.13 | 28.6 | 24.6 | Observed | 1989–1993 | [42] |
| Koga | 287 | 24.3 | - | Rating curve | 2002–2007 | [43] |
| Minchet | 1.13 | 19.3 | 21.8 | Observed | 2010–2014 | [44] |
| Guder | 7011 | 7.5 | - | Rating curve | 2005–2008 | [45] |
| Andasa | 600.6 | 17.9 | 18.1 | Rating curve | 1992–2012 | This study |

### 4.2. Spatial and Temporal Variability of Soil Erosion

Soil erosion spatial variability assessment is crucial in planning and implementation of watershed management strategies [34,50]. Recent studies used the SWAT model to map the spatial variability of sediment yield in various areas of the world [51–54] as well as in Ethiopia [55]. In the Upper Blue Nile basin, studies by Betrie et al. [34], Asres and Awulachew [37], and Ayele et al. [43] assessed spatial variability of sediment yield in varies watersheds. A study by Betrie et al. [34] in the entire Upper Blue Nile basin indicated that from the total 15 sub-basins 8 of them were extremely and severely eroded while 7 sub-basins were moderate and low erosion areas. Most of the extremely and severely eroded areas were located in the steep slope areas, where SWC practices were recommended [34]. The spatial variability of sediment yield in the Gumera watershed conducted by Asres and Awulachew [37] showed that out of the total 30 sub-watersheds, 18 of them produced a mean annual sediment yield ranging 11–22 t ha$^{-1}$ yr$^{-1}$, located in the upstream areas of the watershed. Meanwhile, the bottom slope and wetland areas were characterized by a low sediment yield ranging from 0 to 10 t ha$^{-1}$ yr$^{-1}$ [37].

In Koga watershed, the adjacent watershed to Andasa watershed, Ayele et al. [43] showed that the downstream part of the watershed was severely eroded and prioritized for SWC than the upstream part of the watershed where the area was slightly eroded. The severely eroded sub-watersheds were characterized by Luvisols, cultivated land, and a slope ranging from 2 to 8% [43]. In general, results of the global and local studies demonstrated that steep slope watersheds and cultivated land dominance were recognized as severely eroded areas that require priorities for soil and water conservation (SWC) practices. This study also confirmed that sub-watersheds dominated by cultivated land were identified as erosion hotspot areas. Because of this, it is recommended that the methodology used and similar ones be applied to identify erosion hotspot areas as a planning input for the prioritization of SWC practices. This is crucial to save the natural resources and money on unnecessary investments.

Studies throughout the world agreed that the high soil loss and sediment yield production was directly linked with the records of storms in the wet seasons [56–58]. Similarly, in the case of Ethiopia, a study by Yesuf et al. [59] on the Maybar watershed showed that a high amount of sediment yield was recorded in the month of August when high amounts of rainfall and runoff was recorded. In addition, 1985 and 1981 were the years of high and

low rainfall, runoff and sediment yield, respectively [60]. Similarly to the current study, research by Ebabu et al. [60], Mhiret et al. [61] and Adem et al. [62] also confirmed the association of the amount and intensity of rainfall with soil loss and sediment yield.

## 5. Conclusions

This study used SWAT hydrological and sediment modeling in the Andasa watershed to estimate sediment yield and identify erosion hotspots in areas using a limited amount of data. Despite the various sources of uncertainty, the SWAT model did well in both the calibration and validation periods in predicting streamflow and sediment yield. The study also demonstrated that the SWAT model can generate reliable estimates of the various water balance components. Between 1992 and 2020, the estimated mean annual sediment yield of the Andasa watershed was 17.9 which is within the range of sediment yield estimates for watersheds in the Upper Blue Nile basin. A 132 km$^2$ area in the Andasa watershed that has been severely eroded requires rehabilitation with soil and water conservation (SWC) practices as a top priority. Cultivated land and steep slopes were characteristics of the Andasa watershed's erosion-prone areas. It is advised to collect sediment data more frequently in order to increase the precision of sediment estimation and erosion hotspot area identification. Despite this, the method used in this study is crucial for identification of erosion hotspot areas and the prediction of sediment yield by integrating rating curves and modeling in data-limited regions all over the world.

**Supplementary Materials:** The following supporting information can be downloaded at: https://www.mdpi.com/article/10.3390/hydrology9100167/s1, Table S1. Suspended sediment concentration (SSC) data was used for the discharge-sediment rating curve development, Table S2. Initial streamflow parameters and their range that was used for sensitivity analysis.

**Author Contributions:** Conceptualization B.K.A. and A.A.A.; methodology, B.K.A., A.A.A. and T.G.; software, B.K.A. and A.A.A.; validation, B.K.A. and A.A.A.; formal analysis, B.K.A. and A.A.A.; investigation, B.K.A. and A.A.A.; resources, E.G.D.; data curation, B.K.A. and A.A.A.; writing—original draft preparation, B.K.A.; writing—review and editing, A.A.A. and F.A.Z.; visualization, B.K.A. and A.A.A.; supervision, A.A.A. and K.K.G. All authors have read and agreed to the published version of the manuscript.

**Funding:** This publication was made possible through support provided by the Ministry of Education (MoE) of Ethiopia.

**Data Availability Statement:** The spatial and time series data used in this study was obtained from National Meteorological Agency (NMA) of Ethiopia, Ministry of Water and Energy (MoWE) of Ethiopia, Ethiopian Geospatial Information Center (EGIC) and are available with the permission of the institutions. Sediment concentration data presented in this study are available in Table S1 from the supplementary materials.

**Acknowledgments:** Gondar University and Dilla University provided financial and resources support during the write-up.

**Conflicts of Interest:** The authors declare that do not have a conflict of interest.

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
