# Peer review of "Application of Hydrological and Sediment Modeling with Limited Data in the Abbay (Upper Blue Nile) Basin, Ethiopia"

_hydrology, doi:10.3390/hydrology9100167_

Round 1
Reviewer 1 Report
Interesting work, at a good scientific level. I have a few comments. Regarding Figure 1 (c), the background color of the terrain elevation should be corrected on the map - to those used in geography (the shade of red was incorrectly used to indicate the minimum elevations of the terrain, and the shade of blue was used incorrectly for the maximum elevations).
Due to the fact that the SWAT model underestimates the observed flow rate, and at the same time PBIAS indicated that the model excessively predicts the sludge efficiency, it is necessary to: demonstrate - more broadly prove the correctness of the relationship given in Figure 2, ie Qs = 13.45Q1.90 with R2 = 0.96 (analysis of errors in measurements given in table S1 ?). In turbulent flows of Newtonian fluid, the exponent is 2. Therefore, one should aim at the form: Qs = xQ2 (even with a lower R2 value). Therefore, it would be scientifically interesting to re-attempt the calibration and validation of the model in order to potentially increase its accuracy (?).
Author Response
Dear Reviewer
Thank you for your time and effort in reviewing the manuscript. The comments were extremely helpful in improving the manuscript. Below we have repeated the comments and immediately following it, both our response and the improved text in the revised manuscript. The revised text is in blue.
Thanks again for your time and effort.

Reviewer 2 Report
The manuscript presents a well conducted study with limited data but with interesting results. However, the methodology section related to SWAT model is poorly presented with many missing important information as well as errors, as described hereafter:
· L.150: The methodology on how to calculate most of the given variables in Eq. (1) was not provided. As such, the reader cannot identify the role and place of the main model parameters provided in Table 1.
· What type of data is available for calibration: monthly or daily? How the routing was performed (Tc methods, overland flow equations, …)? The most important notions are missing.
· L.157 to L. 166 describe the original SCS equations and not the modified ones. In fact, the provided SCS equations are for the event based SCS method, which should be modified to account for continuous simulations and calculate soil water content, seepage to vadose zone, return flows….
· As per Eq (2), Qsurf is already in mm not in mm/ha (as mentioned in L.173). Furthermore, units of qpeak in m-3 s-1 seem incorrect.
· It is not clear how equations in section 2.3.2 are used in the procedure described in section 2.3.3. Furthermore, where the relationship developed in Fig 2 is used in the methodology and governing equations?
· Table 1 is very ambiguous: the fitted values are not consistent with commonly acknowledged values (For example CN ranges from 45 to 100, a value of 0.042 is uncommon. As such, the parameter’s definition seems not accurate.)
· Units in Table 1 are unclear. For example, the threshold depth units is not given to assess the value adopted. Fractions mentioned are also not specified.
General comment:
The results could make more use of remote sensing / global datasets for erosion and compare with them to allow for a larger audience to profit from the study results.
Specific comments:
L.118 – L.119: The statement is not correct as the linear equation was fitted to the logarithms or the power equation was fitted to the original data. Please correct.
L.122 and L.123: The website from which data were downloaded is not the most important information; however, the DEM data source and relevant mission is the information to be provided in a scientific paper.
L. 276: The statement starting “the model responds to dual…” is not clear. Please rephrase.
Please remove non-Latin characters from Figure 6.
The results from Figure 11 are expected since the sediment yield was set as a function of discharges and not of rainfall (refer to Eq. 4). Consequently, the discussion should be toned down as the results only reflects the model assumptions.
It is hard to link the discussion in Section 3.5 to Figure 12. Please refer in the text to the watersheds numbers and also add a Figure with Hydrologic Soil Groups to help the reader identify watersheds with type D soil.
Author Response
Dear Reviewer
Thank you for your time and effort in reviewing the manuscript. The comments were extremely helpful in improving the manuscript. Below we have repeated the comments and immediately following it, both our response and the improved text in the revised manuscript. The revised text is in blue.
Thanks again for your time and effort.
6th September 2022

Reviewer 3 Report
The manuscript contains 26% similarity. Authors need to decrease less than 20% before submission.
Author Response
RESPONSE TO THE REVIEW
of the manuscript:
APPLICATION OF HYDROLOGICAL AND SEDIMENT MODELING WITH LIMITED DATA IN THE ABAY (UPPER BLUE NILE) BASIN, ETHIOPIA
Manuscript ID: hydrology-1903106
Dear Reviewer
Thank you for your time and effort in checking the similarity rate of the manuscript. We have put our effort to reduce the overlapping texts with previous studies. However, since the methodology of SWAT hydrological and sediment modeling was adopted from the model user guide and previous literature, the similarity may not be reduced that much. After the first and second reviewers’ comments, we also took our time to revise the manuscript. We are happy to receive your detailed comments in the second round which will be extremely helpful in improving the manuscript.
Thanks again for your time and effort.
6th September 2022
Round 2
Reviewer 3 Report
Current version can be accepted for publication.